# The Effects of Post-Exercise Cold Water Immersion on Neuromuscular Control of Knee

**DOI:** 10.3390/brainsci14060555

**Published:** 2024-05-30

**Authors:** Yuge Wu, Fanjun Qin, Xinyan Zheng

**Affiliations:** School of Exercise and Health, Shanghai University of Sport, Shanghai 200438, China; 2021517007@sus.edu.cn (Y.W.); 40316@shpdh.org (F.Q.)

**Keywords:** cold-water immersion, neuromuscular control, brain activation, muscle activation, knee

## Abstract

To date, most studies examined the effects of cold water immersion (CWI) on neuromuscular control following exercise solely on measuring proprioception, no study explores changes in the brain and muscles. The aim of this study was to investigate the effects of CWI following exercise on knee neuromuscular control capacity, and physiological and perceptual responses. In a crossover control design, fifteen participants performed an exhaustion exercise. Subsequently, they underwent a 10 min recovery intervention, either in the form of passively seated rest (CON) or CWI at 15 °C. The knee proprioception, oxygenated cerebral hemoglobin concentrations (Δ[HbO]), and muscle activation during the proprioception test, physiological and perceptual responses were measured. CWI did not have a significant effect on proprioception at the post-intervention but attenuated the reductions in Δ[HbO] in the primary sensory cortex and posterior parietal cortex (*p* < 0.05). The root mean square of vastus medialis was higher in the CWI compared to the CON. CWI effectively reduced core temperature and mean skin temperature and improved the rating of perceived exertion and thermal sensation. These results indicated that 10 min of CWI at 15 °C post-exercise had no negative effect on the neuromuscular control of the knee joint but could improve subjective perception and decrease body temperature.

## 1. Introduction

Exercise fatigue is a decrease in body functions or exercise intensity resulting from excessive physical activity or continuous competition. It is now known that fatigue can negatively affect power output and performance and increase the risk of sports injuries [1]. A survey of 294 male and 224 female cyclists found that 85% experienced at least one injury caused by exercise fatigue or overuse, with 36% requiring medical treatment [2]. Additionally, the knee joint is the most commonly affected location in cyclists, with 25% experiencing anterior knee pain [3]. Therefore, exploring strategies to alleviate exercise fatigue and promote body recovery is crucial.

Cold water immersion (CWI) has become a popular cryotherapy method for post-exercise recovery, which involves immersing part or all of the body (excluding the head) in cold water (≤20 °C). One review suggests that CWI could enhance recovery of performance in a variety of sports, with immersion in 10–15 °C water for a 5–15 min duration appearing to be most effective at accelerating performance recovery [4]. The combination of hydrostatic pressure and low temperature in CWI may contribute to ameliorating exercise-induced hyperthermia, reduce blood flow and cardiovascular stress, and facilitate the removal of accumulated muscle metabolites [5]. Ascensão et al. [6] reported the beneficial effects of post-exercise CWI on muscular dysfunction and damage in soccer players. They found that CWI (10 min, 10 °C) could significantly reduce markers of muscle damage, accelerate recovery of quadriceps strength, and relieve muscle soreness in various muscle groups. However, some studies suggested that cold exposure might impair neuromuscular control by reducing nerve conduction velocity [7], balance [8], and joint stability during exercise [9], which may increase the risk of injury. While CWI has demonstrated potential benefits for exercise recovery, its application in clinical interventions and exercise practice remains controversial.

The ability to produce controlled movement through coordinated muscle activity is referred to as neuromuscular control, and results from the complex interaction between the nervous and musculoskeletal systems [10]. In a simplified model, the neuromuscular system can be divided into three components: sensory organs, neural pathways, and muscles. Proprioception, an essential aspect of neural control, has been studied to investigate the influence of cooling therapy on neuromuscular control using measurements of joint position reproduction [11,12,13]. Uchio et al. [11] showed that cooling the knee joint with a pad at 4 °C for 15 min increased the stiffness of the anterior terminal of the knee joint and resulted in an increase in knee joint stiffness of 21 N/mm and inaccuracy of position sense of 1.7 degrees. Oliveira et al. [12] compared the effects of cooling applied at different locations on knee position sense. The results showed that cryotherapy impairs the position sense of the knee joint in normal knees, and a similar deleterious effect is observed when cryotherapy is applied over the quadriceps or directly on the knee joint. Chow et al. [13] documented that 1 min of CWI at 5 °C significantly increased the error in knee joint repositioning, while warm water immersion yielded a less deleterious effect on knee joint proprioception than CWI. However, several studies have reported no effect of cryotherapy on joint position [14,15,16,17,18,19]. Costello et al. [18] compared the effects of CWI (14 ± 1 °C) and warm water immersion (28 ± 1 °C) on active ipsilateral limb repositioning sense of the right knee between weight-bearing and non-weight-bearing conditions. The results showed that neither weight-bearing nor non-weight-bearing knee joint position was influenced by cryotherapy. Based on the literature, the effects of cryotherapy, particularly CWI, on knee proprioception are inconsistent. Furthermore, most current studies focused solely on measuring proprioception, without examining the changes in the brain and muscles during the proprioception task after CWI. Therefore, further study is necessary to clarify the effects of CWI on neuromuscular control after exercise-induced fatigue.

The purpose of this study was to investigate the effects of post-exercise CWI on knee neuromuscular control, physiological responses, and subjective perception. We used the knee Active Movement Extent Discrimination Assessment (AMEDA) to measure knee proprioception, functional near-infrared spectroscopy (fNIRS) to monitor brain oxygenation, and surface electromyography (sEMG) to assess muscle activation.

## 2. Materials and Method

### 2.1. Participants

Fifteen healthy and physically active males voluntarily participated in this study, with a mean age of 22.8 ± 2.4 years, height of 176.7 ± 7.5 cm, body mass of 69.7 ± 6.3 kg, and maximum output power (W_max_) of 240 ± 47 W. The sample size used was based on a G*Power 3.1 software calculation (*p* = 80% at α = 0.05; ES = 0.4), and twelve participants were sufficient to minimize the probability of type II error for all the variables. Considering the attrition rate to be 20%, 15 participants were eventually recruited through public social media. The participants were deemed eligible for this study if they met the following criteria: (1) aged between 18 and 23 years old; (2) nonsmoker; (3) exercised at least three times/week; (4) no injury or surgery in the past 6 months. The participants had to abstain from consuming caffeine and alcohol for 12 h before each experiment. In addition, written consent was obtained from all participants prior to the first test. The study was approved (19 May 2022) by the Shanghai University’s Ethical Committee (102772022RT006). 

### 2.2. Experimental Procedures 

During a preliminary laboratory visit, height, body mass, and W_max_ measurements were recorded. W_max_ was determined by an incremental exercise test on a cycle ergometer (Monark 839E, Stockholm, Sweden), starting at 100 W and increasing by 50 W every 3 min until the 9th minute, followed by increments of 25 W per minute until fatigue [20]. Fatigue was defined as participants being unable to maintain a pedaling frequency above 50 rotations per minute (rpm) for more than 30 s despite strong verbal encouragement. 

After the preliminary visit, participants completed two experimental conditions: passive rest and CWI. Randomization was performed using the platform randomization method to avoid potential confusion. The study used a randomized crossover design consisting of two trials with a five-day interval between two trials. The experiments were carried out at the same time of day (±1 h). Participants were instructed to avoid strenuous activity for 48 h and to avoid consuming alcohol and caffeine for 24 h before each trial. To ensure that the physical condition of the participants on the day of the experiment was comparable, they were asked to record their daily activities and meals starting 24 h before the start of the experiment during the familiarization trial and to reproduce the same as much as possible in the follow-up experiments.

Upon arrival for each experimental trial, participants performed a 5 min warm-up on a cycle ergometer followed by an exhaustion exercise, which involved cycling at 40% of W_max_ for 30 s, followed by 120% of W_max_ for 30 s, repeated until participants could not maintain a single 30 s period [20]. Cycling time to fatigue was recorded. After the exercise, a 10 min intervention break was provided, during which participants were either immersed in cold water (CWI) or remained seated in the laboratory (CON). During the CWI, the participants were submerged in an inflatable pool with their neck and head out of the water. The water temperature was maintained at 15 °C by a specially designed water refrigeration unit (iCool Portacovery, Gold Coast, Australia). In the present study, the water temperature was chosen in accordance with Brophy-Williams et al. [21].

### 2.3. Measurements

Knee proprioception was assessed using the AMEDA test, which incorporates various sensory information such as texture, vision, and hearing, closely resembling real-world conditions and suitable for studying the role of proprioception in practical applications [22]. Each trial of the AMEDA test consisted of four movement displacement distances (from the smallest, position 1, to the largest, position 4), with each position occurring randomly 10 times, yielding a total of 40 movements. Participants underwent a familiarization session to become acquainted with the testing procedure before data collection commenced. They were instructed to judge and record the position number of each test movement without receiving feedback on the accuracy of their judgments for each trial. The area under the ROC curve (AUC) was calculated as an index of participants’ ability to distinguish between two joint movements. The test was conducted pre- and post-exercise, post-intervention, and 24 h post-exercise.

Muscle electrical activity was recorded during the AMEDA test using differential surface electrodes (Myon AG, Schwarzenberg, Switzerland). The electrodes were positioned over specific muscles on the right thigh, including rectus femoris (RF), vastus medialis (VM), vastus lateralis (VL), biceps femoris (BF), semitendinosus (SE), gastrocnemius medialis (GM), and gastrocnemius lateralis (GL). Electrode placements were marked on the skin to ensure consistent repositioning 24 h post-exercise. Before applying the electrodes, participants’ skin was shaved, rubbed, and cleaned with alcohol. The electrodes and cables were securely taped to minimize movement artifacts. Data collection and analysis were performed using EMG and Motion Tools Version 8.7.6.0 computer software. Root mean square (RMS) and median frequency (MF) were recorded. The sEMG data were sampled at a rate of 2000 Hz and band-pass filtered between 10 and 500 Hz. The processed data were equalized and analyzed using the root mean square method with a 50 millisecond time constant. The RMS was calculated for each muscle during the 3 s knee flexion period, then averaged and normalized to the maximal voluntary isometric contraction. For frequency domain data, a fast Fourier transform was applied to the sEMG signals to calculate the MF. 

Changes in oxygenated cerebral hemoglobin concentrations (Δ[HbO]) during the AMEDA test were examined using an fNIRS instrument (NIRSport2, NIRx Medical Technologies LLC, Glen Head, NY, USA). The NIRS probe consisted of 8 sources and 7 detectors placed over the left precentral gyrus to the superior parietal area, between FCZ and POZ (based on the international EEG 10–20 system). The probe was divided into three regions of interest (ROI): ROI 1 represented the primary motor cortex (M1), with channels 11, 12, and 14–21; ROI 2 represented the primary sensory cortex (S1), with channels 7–11 and 13–14; and ROI 3 represented the posterior parietal cortex (PPC), with channels 1–8 and 22–23 (Figure 1). The inter-optode distance was set at 45 mm using a transparent plastic spacer, and a black elastic headband was worn over the probe to secure placement and minimize the effects of ambient light. The fNIRS signals were first converted into optical density and then processed with a high-pass filter at 0.01 Hz to remove baseline drift and low-frequency oscillations and a low-pass filter at 0.2 Hz to reduce the effects of cardiovascular artifacts and high-frequency noise. Δ[HbO] for each knee flexion during the AMEDA test was averaged to represent changes in brain activation. Raw light intensity was converted into changes in hemoglobin concentration using the modified Beer–Lambert law.

Physiological measurements, including heart rate (HR) and skin temperature, were recorded every 2 min during exercise and the intervention period. HR was monitored using a chest strap and a wristwatch receiver (model RS400; Polar Electro Oy, Kemple, Finland). Skin temperature was measured using a wireless thermistor probe (BODYCAP e-Tact; BODYCAP S.A.S., Normandie, France), which was attached to the skin surface of the upper arm (T_arm_), chest (T_chest_), and thigh (T_thigh_). Data were monitored using eTact Watcher software (BODYCAP S.A.S., France). The mean skin temperature (T_skin_) was calculated using the formula [23]: T_skin_ = 0.25 × T_arm_ + 0.43 × T_chest_ + 0.32 × T_thigh_. Core temperature (T_core_) was measured with an ingestible temperature sensor capsule (BODYCAP P022-P; BODYCAP S.A.S., Normandie, France) and recorded with a telemetric data receiver (BODYCAP e-Viewer Performance; BODYCAP S.A.S., Normandie, France). To ensure accurate data collection during each trial, participants were required to ingest the core temperature capsule orally two hours before exercise. 

Perceptual index included the rating of perceived exertion (RPE: 6 = Very, Very Light; 20 = Very, Very Hard) [24], perceived thermal sensation (TS, 0 = unbearably cold—8 = unbearably hot) [25], and perceived muscle soreness (MS, 0 = normal—10 = extremely sore) [26]. RPE, TS, and MS were measured pre- and post-exercise, post-intervention, and 24 h post-exercise. 

### 2.4. Statistical Analysis

Statistical analyses were performed using SPSS 21.00. Data were checked for normality and variance using the Kolmogorov–Smirnov test. A paired t test was used to evaluate differences in cycling time to fatigue. A two-way (condition × time) repeated measures analysis of variance (ANOVA) was conducted to determine differences between cooling conditions (CWI vs. CON). Repeated measures ANOVA (pre- vs. post- vs. 24 h post-exercise vs. post-intervention) was employed to study the effects of the intervention. Post hoc analyses with Bonferroni correction were performed when interaction effects were identified. The effect size was measured using partial eta-squared (η_p_^2^), with the following criteria for interpretation: small (η_p_^2^ = 0.01), medium (η_p_^2^ = 0.06), and large (η_p_^2^ = 0.14). All data were reported as mean ± SD. A *p*-value < 0.05 was statistically significant.

## 3. Results

There was no significant difference in cycling time to fatigue between CON and CWI groups (CON, 16 ± 5.5 min; CWI, 16 ± 17.6 min, *p* = 1.00), suggesting that the physical conditions for the two groups were comparable. 

Regarding the AMEDA test, a 2 × 4 mixed ANOVA revealed no significant interaction (F (3,26) = 0.100, *p* = 0.959, η_p_^2^ = 0.011, Figure 2). However, significant main effects were observed for the time factor (F (3) = 26.450, *p* = 0.000, η_p_^2^ = 0.753). The AUC was significantly reduced from pre- to post-exercise in both the CON group (pre-exercise: 0.87 ± 0.06, post-exercise: 0.82 ± 0.05, *p* < 0.001) and the CWI group (pre-exercise: 0.86 ± 0.05, post-exercise: 0.82 ± 0.05, *p* < 0.001), without significant differences between two groups (*p* > 0.05). After the intervention, proprioception in the CON group remained significantly worse than pre-exercise (pre-exercise: 0.87 ± 0.06, post-intervention: 0.82 ± 0.08, *p* < 0.05), but no significant differences were observed in the CWI group (CWI, pre-exercise: 0.86 ± 0.05, post-intervention: 0.83 ± 0.05, *p* > 0.05). 

For ROI 1, a 2 × 4 mixed ANOVA revealed that there was no significant interaction of Δ[HbO] (F (3,84) = 0.062, *p* = 0.980, η_p_^2^ = 0.002, Figure 3A), but there were significant main effects for the time (F (3) = 4.093, *p* = 0.009, η_p_^2^ = 0.128). In the CON group, Δ[HbO] significantly decreased at the post-intervention and 24 h post-exercise time points compared to post-exercise (*p* < 0.05). For ROI 2, a 2 × 4 mixed ANOVA revealed no significant interaction (F (3,84) = 1.081, *p* = 0.362, η_p_^2^ = 0.037, Figure 3B), but there were significant main effects for the time (F (3) = 3.148, *p* = 0.042, η_p_^2^ = 0.266). After the intervention, Δ[HbO] in the CON group significantly decreased compared to pre- and post-exercise (*p* < 0.05) and was lower than in the CWI group (*p* < 0.05). For ROI 3, a 2 × 4 mixed ANOVA revealed no significant interaction (F (3,84) = 1.696, *p* = 0.174, η_p_^2^ = 0.057, Figure 3C), but there were significant main effects for the time (F (3) = 4.144, *p* = 0.009, η_p_^2^ = 0.129). After the intervention, Δ[HbO] in the CON group decreased significantly compared to pre-exercise (*p* < 0.05) and was lower than in the CWI group (*p* < 0.05). 

For RMS, a 2 × 4 mixed ANOVA revealed no significant interaction or main effects for RF, VM, VL, BF, and GL (Table 1). There was also no significant interaction or main effects for SE and GM (SE: F_time_ (3) = 1.642, *p* = 0.186, η_p_^2^ = 0.055, F_inter_ (3,84) = 0.292, *p* = 0.831, η_p_^2^ = 0.010; GM: F_time_ (3) = 0.245, *p* = 0.865, η_p_^2^ = 0.009, F_inter_ (3,84) = 0.666, *p* = 0.576, η_p_^2^ = 0.023). After exercise, the RMS of RF increased significantly in both groups (*p* < 0.05). The RMS of VM in the CWI group was higher than that in the CON group at the post-intervention (*p* < 0.05) and increased significantly immediately after the intervention and 24 h compared to baseline (*p* < 0.01). Regarding MF, a 2 × 4 mixed ANOVA revealed no significant interaction for VM (F (3,84) = 0.338, *p* = 0.798, η_p_^2^ = 0.012), but there were significant main effects for time (F (3) = 6.756, *p* = 0.000, η_p_^2^ = 0.194). There were no significant interactions or main effects for RF, VL, SE, BF, GM, and GL. After exercise, MF decreased significantly in both groups (CON: *p* = 0.053, CWI: *p* < 0.05). 

For T_core_, a 2 × 4 mixed ANOVA revealed a significant interaction (F (3,84) = 17.953, *p* = 0.000, η_p_^2^ = 0.391, Figure 4A). T_core_ in the CWI group decreased significantly during the proprioception test (*p* < 0.05) and was lower than that in the CON group (*p* < 0.01). For T_skin_, a 2 × 4 mixed ANOVA revealed a significant interaction (F (3,26) = 58.367, *p* = 0.000, η_p_^2^ = 0.871, Figure 4B). After the intervention, T_skin_ significantly decreased in both the CON group (*p* < 0.05) and the CWI group (*p* < 0.01), with T_skin_ being lower in the CWI group than in the CON group (*p* < 0.01). 

For HR, a 2 × 8 mixed ANOVA revealed a significant interaction (F (7,22) = 4.255, *p* = 0.004, η_p_^2^ = 0.575, Figure 5). In the CON group, HR increased significantly until the 8th minute during the intervention compared to the pre-exercise (*p* < 0.01). In the CWI group, HR increased until the 2nd minute during immersion. Moreover, during the 0–8 min interval of the intervention, HR was significantly higher in the CON group than in the CWI group (*p* < 0.05).

For RPE, a 2 × 4 mixed ANOVA revealed a significant interaction (F (3,84) = 4.703, *p* = 0.004, η_p_^2^ = 0.144, Figure 6A). After the intervention, RPE in the CON group still increased significantly compared to pre-exercise (*p* < 0.05) and was higher than in the CWI group (*p* < 0.01). For MS, a 2 × 4 mixed ANOVA revealed no significant interaction (F (3,84) = 2.168, *p* = 0.098, η_p_^2^ = 0.072, Figure 6B), but there were significant main effects for time (F = 149.896, *p* = 0.000, η_p_^2^ = 0.843). After exercise, MS increased significantly in both groups (*p* < 0.01). After the intervention, MS in the CON group remained elevated compared to pre-exercise (*p* < 0.01) and higher than that in the CWI group (*p* < 0.05).

For TS, a 2 × 9 mixed ANOVA revealed a significant interaction (F (8,21) = 10.158, *p* = 0.000, η_p_^2^ = 0.795, Figure 7). During the intervention, TS in the CWI group remained lower than that in the CON group (*p* < 0.01) and decreased significantly compared to pre-exercise (*p* < 0.05).

## 4. Discussion

To our knowledge, this is the first study to investigate the effects of CWI after fatigue on neuromuscular control of knee joints and its physiological and perceptual parameters. In this study, CWI (15 °C, 10 min) was effective in decreasing the T_core_, T_skin_, HR, TS, RPE, and MS, as well as inhibiting the impairment of the oxyhemoglobin concentration in S1 and PPC, RMS of VM. In addition, post-exercise CWI had no effect on knee proprioception, oxyhemoglobin concentration in M1, and muscle activation of RF, VL, BF, SE, GM, and GL. 

Neuromuscular control can be simplified into three components: sensory, neural pathways, and muscles [10]. To date, although several studies have examined the effects of post-exercise cryotherapy on neuromuscular function, most have focused only on the effects on proprioception, and the effects remain controversial [10,13,18,19]. Uchio et al. [11] found that 15 min of cooling at 4 °C increased knee stiffness (21 N/mm) and decreased the ability to accurately reproduce the target angle. Chow et al. [13] observed similar effects and found that 1 min of CWI at 5 °C significantly increased the error in repositioning the knee joint. However, Khanmohammadi et al. [19] reported that CWI had no negative impact on proprioception, either immediately or 24 h post-exercise. In this study, our results were supported by the findings of Costello et al. [18] and Khanmohammadi et al. [19], suggesting that CWI has no negative impact on proprioception, either immediately or 24 h post-exercise. The inconsistent effects of CWI on proprioception could be related to the degree of cooling. It is well known that proprioception is influenced by deep tissue temperature and nerve conduction. Skin temperature is a key parameter affecting nerve conduction velocity, and cold exposure reduces neuromuscular conduction velocity. When skin temperature decreases by 12.5 °C, nerve conduction velocity decreases by 10% [27]. However, the 10 min CWI at 15 °C used in this study only reduced T_skin_ by 4.2 °C, resulting in a reduction in nerve conduction velocity of 3% or less, which may not be sufficient to impair proprioception. 

Previous studies have shown that the cerebral regions responsible for proprioceptive performance are located in M1, S1, PPC, insula, and cerebellum [28,29]. In this study, we observed that exercise-induced fatigue significantly reduced knee proprioception compared to the pre-exercise, but there was an increasing trend in oxyhemoglobin concentration in M1, S1, and PPC. The increase in oxygenated hemoglobin in these regions may result in preventing further decline in proprioception after exercise. Exercise fatigue leads to neuromuscular changes, which in turn reduce a person’s ability to control and dynamically stabilize the lower limbs. To prevent further decline in proprioception after exercise, more blood oxygen needs to be mobilized to improve brain activation. Furthermore, in the present study, we found that the change in oxyhemoglobin concentration in S1 and PPC was significantly higher in the CWI group, along with lower TS in the CON group after the intervention, but there was no significant difference in the proprioception between the CON and CWI groups. Therefore, the higher change in oxyhemoglobin concentration in S1 and PPC in the CWI group after the intervention may be related to the activation of brain regions associated with thermal sensation (S1 and PPC).

It has been reported that exercise-induced fatigue leads to an increase in RMS and a decrease in the MF of muscles [30]. Consistently, in this study, the RMS value of RF increased significantly in both groups after exercise compared to pre-exercise, and the RMS of other muscles also showed a tendency to increase. The increased RMS after exercise may be due to the recruitment of more muscle fibers to maintain continued performance in the AMEDA test after fatigue. Additionally, RMS reflects the effectiveness of muscle firing, which is related to the synchronization of excitatory rhythms and the recruitment of motor units. In this study, we found that after the intervention, the RMS of VM was significantly higher in the CWI group, along with a similar trend in other muscles compared to the CON group. These results are supported by the results of the previous study, in which RMS increased to different extents after CWI [31]. The increase in RMS after CWI could be related to the increased recruitment of fast-twitch fibers. Different temperature stimuli can have a significant impact on muscle recruitment. When exposed to high-temperature environmental stimulation, fast-twitch fibers are recruited later than slow-twitch fibers, whereas fast-twitch fibers are recruited earlier when exposed to low-temperature environmental stimulation [32]. Therefore, due to the different proportions of muscle fiber distribution in the superficial area of each muscle, the RMS of the different muscles showed varying degrees of increase. 

In the present study, a 10 min CWI at 15 °C decreased T_skin_ by about 4.2 °C. Meanwhile, TS was significantly lower in the CWI group compared to the CON group. It has been suggested that T_core_ and T_skin_ influence TS both at rest and during exercise. Compared to the almost constant T_core_, T_skin_ had a greater effect on TS and these two variables were positively correlated [30]. Therefore, the decrease in TS after CWI may be related to the decrease in T_skin_. Furthermore, a systematic review with meta-analysis and meta-regression suggests that CWI was an effective recovery tool after high-intensity exercise, with positive outcomes occurring for muscular power and perceived recovery [33]. McCarthy et al. [34] reported that 10 min of CWI at 8 °C significantly reduced T_core_, HR, and RPE during subsequent exercise and improved subsequent endurance exercise performance. Consistently, we found that CWI could effectively reduce T_core_, HR, and RPE. Although we did not measure post-intervention physical performance in this study, we speculate that 10 min of CWI at 15 °C also has the potential to aid recovery. The reduction in T_core_ via CWI could increase global electroencephalographic β activity (and presumably overall α:β ratio) and decrease a sense of RPE. Therefore, CWI closely mirrors the extent of central fatigue during exercise, resulting in a more even pacing strategy so that higher power outputs can be better maintained during the subsequent exercise [35]. Considering that 10 min of CWI at 15 °C did not affect neuromuscular control or cause joint injury during subsequent exercise, it can be concluded that 10 min of CWI at 15 °C can be used after training or between competitions to promote recovery. Moreover, it is important to note that decreasing RPE through CWI may lead to a false perception of reduced fatigue, which in turn may induce overwork.

## 5. Conclusions

Compared to passive recovery, 10 min of CWI at 15 °C post-exercise reduced HR, T_skin_, and T_core_, and improved TS, RPE, and MS. Furthermore, post-exercise CWI had no effect on proprioception but enhanced brain oxygenation in S1 and PPC as well as the RMS of VM. These results indicate that 10 min of CWI at 15 °C post-exercise could improve subjective perception and reduce body temperature but yielded no negative effect on the neuromuscular control of the knee joint. Based on our findings, we suggest that 10 min of CWI at 15 °C can be used after training or between competitions to promote recovery.

## Figures and Tables

**Figure 1 brainsci-14-00555-f001:**
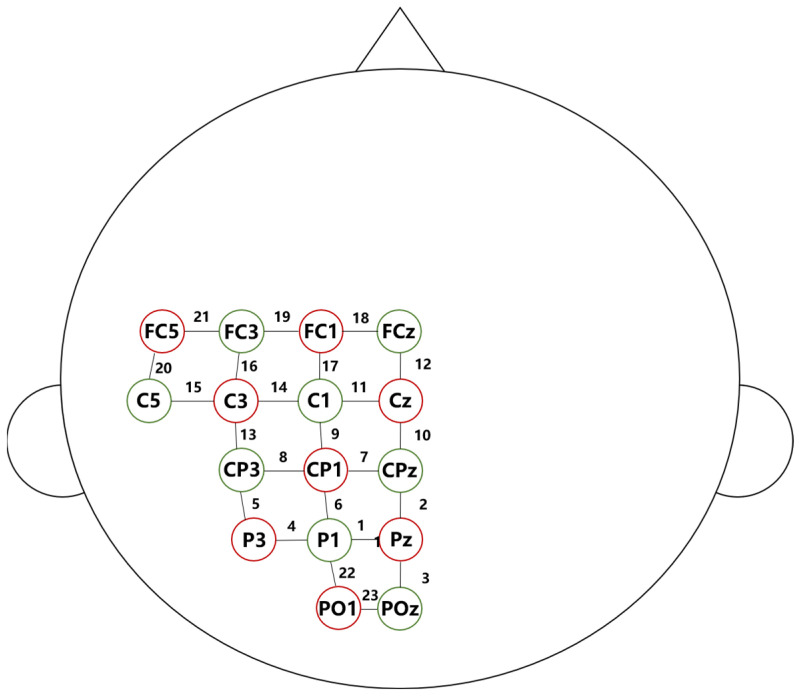
The spatial profile of functional near-infrared spectral imaging (fNIRS) probes. The red circles indicate the 8 optical sources, the green circles indicate the 7 detectors and the black numbers (1–23) indicate fNIRS channels. The optical sources and detectors were positioned on the international 10–20 standard positions.

**Figure 2 brainsci-14-00555-f002:**
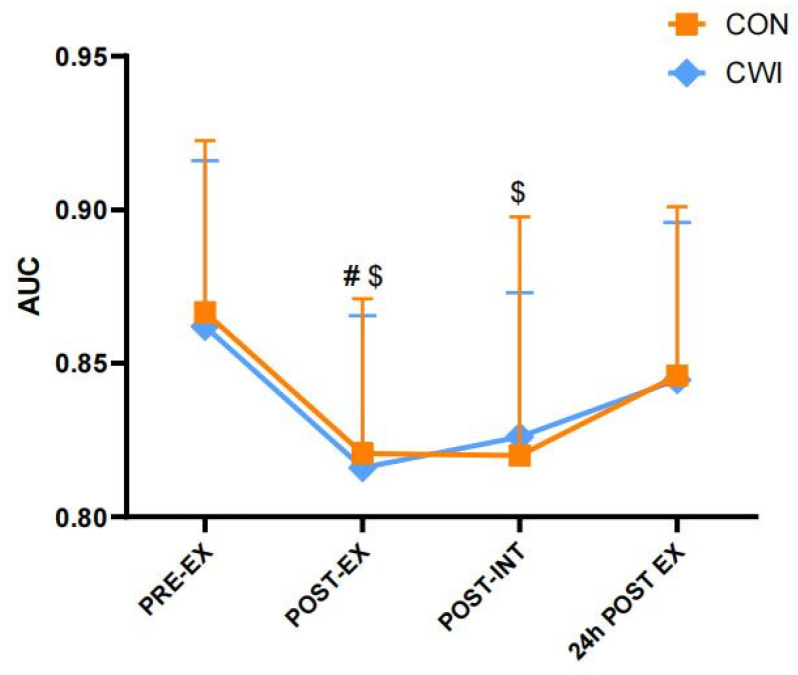
Changes in knee proprioception for control (CON) and cold-water immersion (CWI) conditions. AUC, the area under the receiver operating characteristic curve of the knee Active Movement Extent Discrimination Assessment test. PRE-EX, pre-exercise; POST-EX, post-exercise; POST-INT, post-intervention; 24 h POST EX, 24 h post-exercise. # significantly different compared with pre-exercise in CWI. $ significantly different compared with pre-exercise in CON. The values are shown as mean ± SD. *p* < 0.05 was considered statistically significant.

**Figure 3 brainsci-14-00555-f003:**
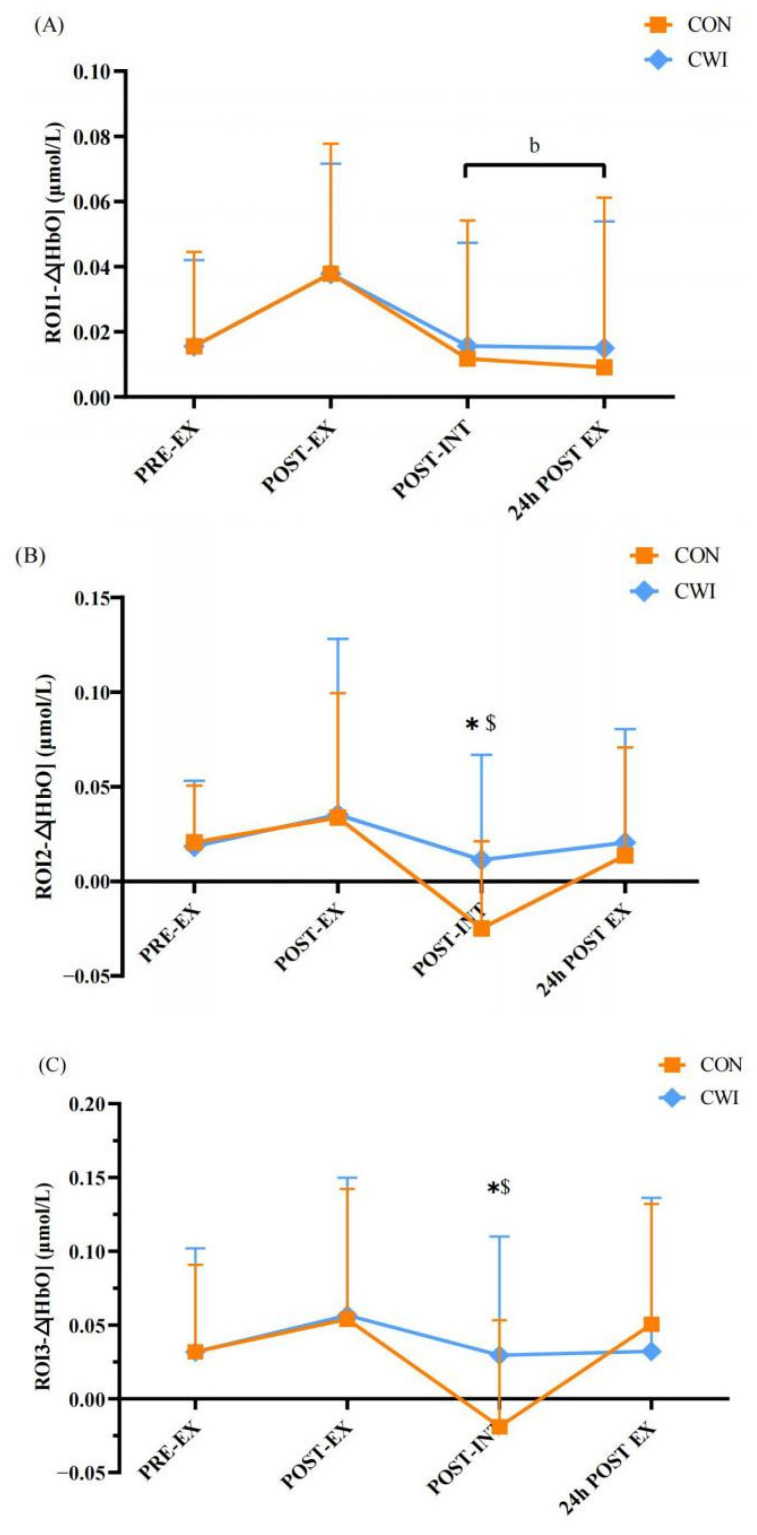
Changes in the primary motor cortex (ROI1, (**A**)), the primary sensory cortex (ROI2, (**B**)), and the posterior parietal cortex (ROI3, (**C**)) for control (CON) and cold-water immersion (CWI) conditions. PRE-EX, before exercise; POST-EX, after exercise; POST-INT, after intervention; 24 h POST EX, 24 h after exercise. * significantly different compared to CON. b significantly different compared to post-exercise in CON. $ significantly different compared to pre-exercise in CON. The values are shown as mean ± SD. *p* < 0.05 was considered statistically significant.

**Figure 4 brainsci-14-00555-f004:**
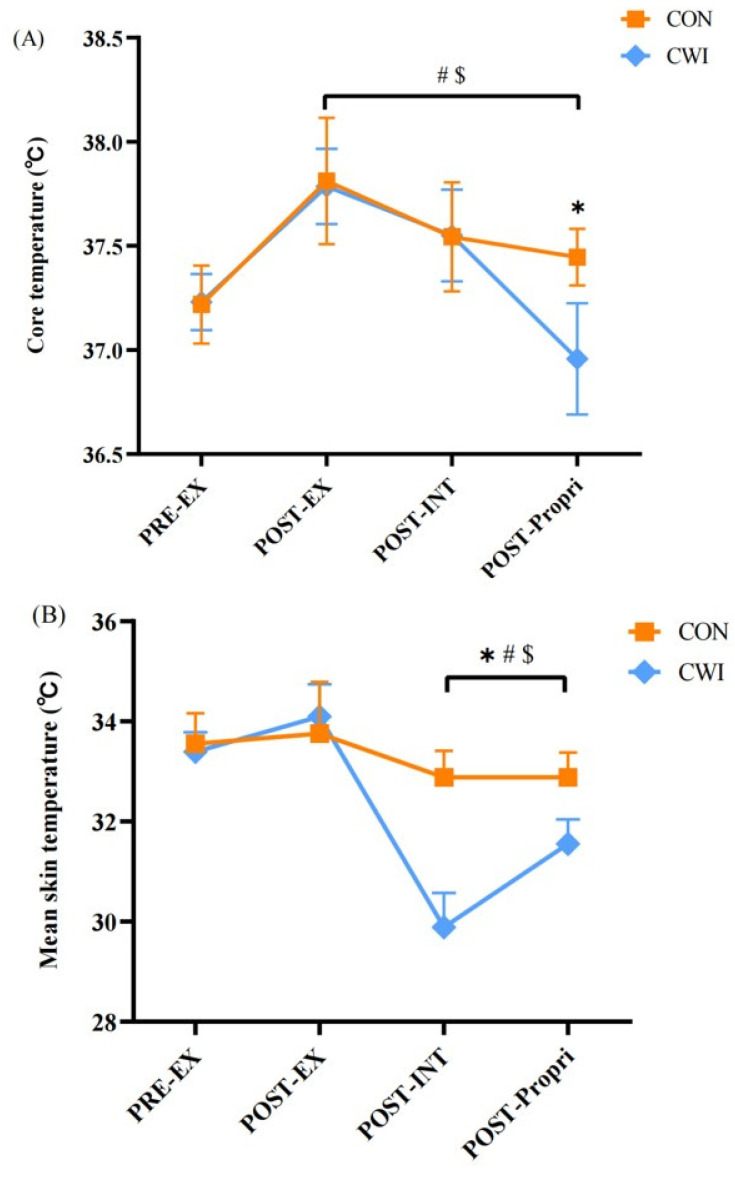
Change in core temperature (**A**) and mean skin temperature (**B**) for control (CON) and cold-water immersion (CWI) conditions. PRE-EX, before exercise; POST-EX, after exercise; POST-INT, after intervention; POST-Propri, the proprioception test after intervention; 0–10, during the intervention time. * significantly different compared to CON. # significantly different compared to pre-exercise in CWI. $ significantly different compared to pre-exercise in CON. The values are shown as mean ± SD. *p* < 0.05 was considered statistically significant.

**Figure 5 brainsci-14-00555-f005:**
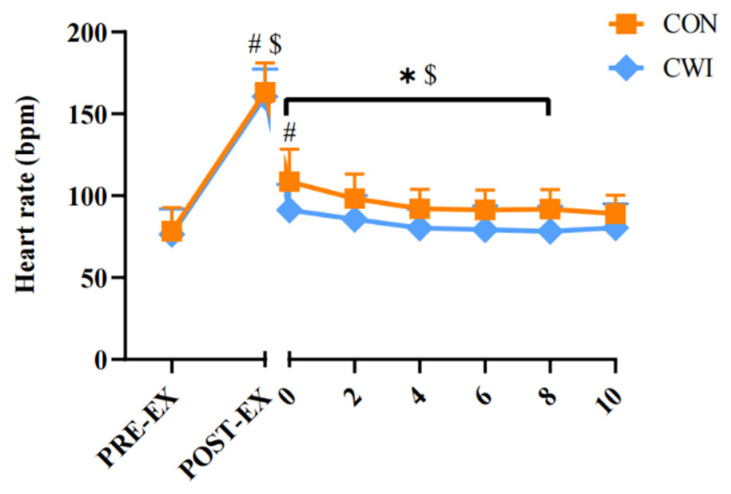
Change in heart rate for control (CON) and cold-water immersion (CWI) conditions. PRE-EX, before exercise; POST-EX, after exercise; POST-INT, after intervention; POST-Propri, the proprioception test after intervention; 0–10, during the intervention time. * significantly different compared to CON. # significantly different compared to pre-exercise in CWI. $ significantly different compared to pre-exercise in CON. The values are shown as mean ± SD. *p* < 0.05 was considered statistically significant.

**Figure 6 brainsci-14-00555-f006:**
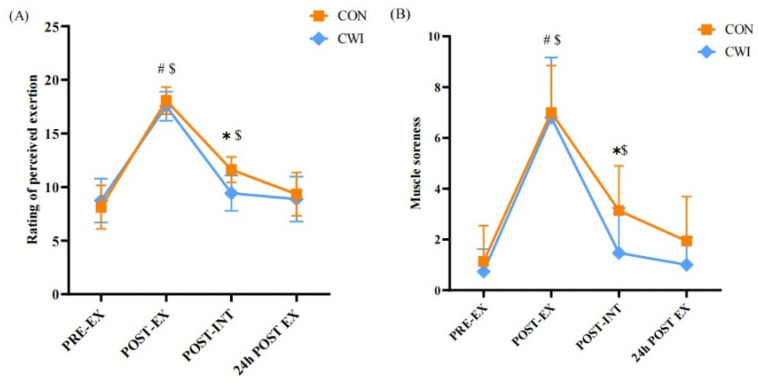
Change in rating of perceived exertion (**A**) and muscle soreness (**B**) for control (CON) and cold-water immersion (CWI) conditions. PRE-EX, before exercise; POST-EX, after exercise; POST-INT, after intervention; 24 h POST EX, 24 h after exercise; 0–10, during the intervention time. * significantly different compared to CON. # significantly different compared to pre-exercise in CWI. $ significantly different compared to pre-exercise in CON. The values are shown as mean ± SD. *p* < 0.05 was considered statistically significant.

**Figure 7 brainsci-14-00555-f007:**
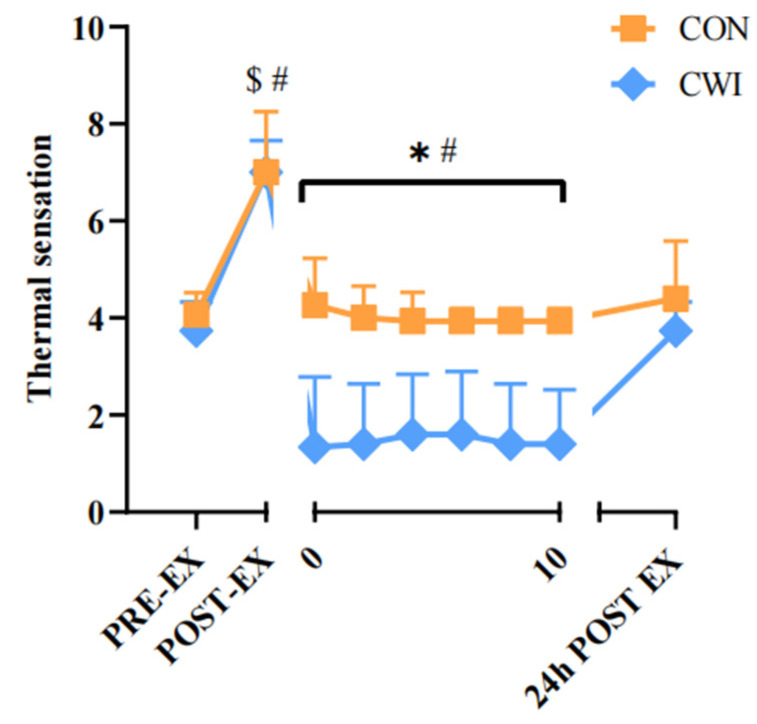
Change in thermal sensation for control (CON) and cold-water immersion (CWI) conditions. PRE-EX, before exercise; POST-EX, after exercise; POST-INT, after intervention; 24 h POST EX, 24 h after exercise; 0–10, during the intervention time. * significantly different compared to CON. # significantly different compared to pre-exercise in CWI. $ significantly different compared to pre-exercise in CON. The values are shown as mean ± SD. *p* < 0.05 was considered statistically significant.

**Table 1 brainsci-14-00555-t001:** Electrical muscle activity of the knee.

Muscle	Time	RMS (μV)		MF (Hz)	
		CON	CWI	CON	CWI
RF	PRE-EX	3.7 ± 2.0	3.8 ± 1.8	70.0 ± 3.9	70.0 ± 4.1
	POST-EX	6.1 ± 2.9 ^$^	6.1 ± 3.3 ^#^	68.7 ± 7.0	67.6 ± 5.1
	POST-INT	4.8 ± 2.6	5.4 ± 2.6	67.9 ± 9.0	66.6 ± 8.0
	24 h POST EX	5.6 ± 2.7	5.5 ± 3.0	69.5 ± 5.0	69.9 ± 6.7
VM	PRE-EX	6.9 ± 3.3	6.8 ± 3.3	70.9 ± 5.9	71.2 ± 6.5
	POST-EX	8.8 ± 3.8	8.9 ± 3.9	67.4 ± 5.9 ^$^	66.1 ± 4.0 ^#^
	POST-INT	10.0 ± 4.3	14.1 ± 7.8 *^#^	69.7 ± 5.3	69.7 ± 5.4
	24 h POST EX	10.7 ± 5.8	14.0 ± 3.8 ^#^	70.7 ± 4.7	71.4 ± 6.2
VL	PRE-EX	10.8 ± 3.4	11.4 ± 6.0	72.7 ± 5.3	72.9 ± 6.9
	POST-EX	12.0 ± 6.2	12.0 ± 7.0	72.5 ± 4.5	71.9 ± 4.8
	POST-INT	14.1 ± 7.6	19.5 ± 3.4	73.9 ± 6.2	71.2 ± 8.9
	24 h POST EX	14.0 ± 6.4	19.0 ± 3.3	72.6 ± 10.3	71.2 ± 8.7
SE	PRE-EX	3.9 ± 2.7	3.9 ± 2.3	72.0 ± 2.9	72.3 ± 2.4
	POST-EX	4.9 ± 3.7	4.9 ± 3.2	70.8 ± 5.9	71.4 ± 8.0
	POST-INT	3.3 ± 1.8	3.9 ± 3.1	72.0 ± 5.0	71.3 ± 8.6
	24 h POST EX	4.6 ± 2.9	4.0 ± 3.7	70.2 ± 5.3	71.8 ± 5.3
BF	PRE-EX	11.3 ± 6.1	11.5 ± 6.1	75.9 ± 6.2	75.4 ± 7.0
	POST-EX	15.4 ± 6.0	14.5 ± 3.3	72.8 ± 5.9	73.6 ± 6.3
	POST-INT	12.9 ± 6.6	13.2 ± 5.5	72.3 ± 5.9	72.1 ± 10.3
	24 h POST EX	11.9 ± 3.6	13.7 ± 6.4	73.0 ± 8.5	75.1 ± 7.2
GM	PRE-EX	12.9 ± 6.6	12.8 ± 5.9	77.6 ± 3.8	77.1 ± 4.1
	POST-EX	13.7 ± 8.6	14.1 ± 4.1	76.3 ± 6.7	76.8 ± 6.2
	POST-INT	12.6 ± 4.2	14.7 ± 4.2	78.2 ± 4.8	75.9 ± 6.8
	24 h POST EX	14.0 ± 6.0	12.2 ± 5.7	80.2 ± 6.9	77.7 ± 7.9
GL	PRE-EX	17.1 ± 5.1	16.9 ± 5.3	78.0 ± 4.2	77.3 ± 5.4
	POST-EX	20.5 ± 7.0	21.0 ± 7.8	77.3 ± 9.0	76.6 ± 6.4
	POST-INT	21.1 ± 6.0	21.0 ± 5.6	78.4 ± 5.7	75.3 ± 6.3
	24 h POST EX	22.3 ± 6.0	23.1 ± 7.3	79.0 ± 7.5	77.9 ± 6.3

PRE-EX, before exercise; POST-EX, after exercise; POST-INT, after intervention; 24 h POST EX, 24 h after exercise. Muscle: RF, rectus femoris; VM, medial vastus; VL, lateral vastus; SE, semitendinosus; BF, biceps femoris; GM, gastrocnemius medialis; GL, gastrocnemius lateralis. * significantly different compared to CON (*p* < 0.05). # significantly different compared to pre-exercise in CWI (*p* < 0.05). $ significantly different compared to pre-exercise in CON (*p* < 0.05).

## Data Availability

The original data presented in the study are openly available in [FigShare] atURL [10.6084/m9.figshare.23896521] (14 April 2024).

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
