# Peer review of "The Effects of Post-Exercise Cold Water Immersion on Neuromuscular Control of Knee"

_brainsci, 2024, doi:10.3390/brainsci14060555_

Round 1

Reviewer 1 Report

Comments and Suggestions for Authors

Manuscript ID: brainsci-2987042

The Effects of Post-Exercise Cold Water Immersion on Neuromuscular

Control of Knee

One Inclusion criterion was exercise at least three times a week. Neither the type nor the extent of exercise were defined. The 15 subjects might  have been in entirely different physical conditions

15 degrees is not really cold water

The cohort investigated may have not been homogenous with regard to physical condition

4/24

Comments on the Quality of English Language

no

Author Response

       We appreciate the reviewer’s suggestions, which will help to increase the scientific quality of the paper and improve the resubmitted version. The corrections according to the comments in the revised manuscript are marked in red and underline. We will answer in a point-to-point fashion to the comments of the reviewer.

One Inclusion criterion was exercise at least three times a week. Neither the type nor the extent of exercise were defined. The 15 subjects might  have been in entirely different physical conditions.

Thanks for your comments.  In this study, we recruited tea-sports (soccer, basketball, rugby) collegiate male players as subjects. The reason that one inclusion criterion was exercise at least three times a week is to ensure the participants has ability to complete high intensity exercise protocol. To avoid different physical conditions, the randomized crossover within-subjects design was used in the present study. Moreover, participants were instructed to avoid strenuous activity for 48 hours before each trial and to refrain from consuming alcohol and caffeine for 24 hours. To ensure that the physical status of the participants was comparable on the day of the experiment, they were asked to record their daily activities and meals from 24 hours prior to the commencement of the experiment during the familiarization trial and to reproduce the same as much as possible in the subsequent experiments. We have added these words in our manuscript.

15 degrees is not really cold water

Thanks for your comments. Tipton et  al. (2018) reported that there is no strict definition of ‘cold water’. In laboratory conditions, the respiratory frequency response (an indication of respiratory drive) peaks on naked immersion in a water temperature between 15°C and 10°C, and is no greater on immersion in water at 5°C (Tipton et al. 1991). These results indicated that some of the hazardous responses to cold water appear to peak on immersion somewhere between 15°C and 10°C. A review reported that water immersion has been divided into four techniques according to water temperature: cold water immersion (CWI; ≤20 °C), hot water immersion (HWI; ≥36 °C), contrast water therapy (CWT; alternating CWI and HWI) and thermoneutral water immersion (TWI; >20 to <36 °C) (Versey et al., 2013). Thus, 15°C can be defined as cold water. Moreover, previous studies have been used 15°C as cold water immersion for body cooling.  Therefore, the water temperature in the present study was chosen in accordance with Vaile et al. [21] and Peiffer et al. [22]. We have added these words in our manuscript.

The cohort investigated may have not been homogenous with regard to physical condition

Thanks for your comments. To ensure that the physical status of the participants was comparable on the day of the experiment, they were asked to record their daily activities and meals from 24 hours prior to the commencement of the experiment during the familiarization trial and to reproduce the same as much as possible in the subsequent experiments. We have added these words in our manuscript. Participants were instructed to avoid strenuous activity for 48 hours before each trial and to refrain from consuming alcohol and caffeine for 24 hours. Moreover, cycling time to fatigue was recorded to evaluating participants’ physical conditions. There was no significant difference in cycling time to fatigue between CON and CWI groups (CON, 16 ± 5.5 min; CWI, 16 ± 17.6 min, p = 1.00), indicating that the physical conditions for the two groups were comparable.

Reviewer 2 Report

Comments and Suggestions for Authors

The authors demonstrated in this paper that CWI at 15°C for 10 minutes after exercise may promote recovery from post-exercise fatigue. The reviewer found it particularly significant that the authors used fNIRS to show changes in ΔOHb in ROIs 1 and 2.

The reviewer have the following concerns and would appreciate additional clarification.

1. about CWI

Please explain the rationale for selecting the protocol of 15°C for 10 minutes.

2. interpretation of the effect of CWI

The reviewer agrees that CWI reduced the perception of unpleasant sensory stimuli associated with exercise-induced fatigue, but the reviewer has the impression that it is difficult to argue about fatigue recovery from this study. Rather, the reviewer is concerned that the reduction of unpleasant sensory stimuli may induce the false perception that fatigue is reduced, which in turn may induce overwork. Please clarify this point in the Discussion.

Author Response

We appreciate the reviewer’s suggestions, which will help to increase the scientific quality of the paper and improve the resubmitted version. The corrections according to the comments in the revised manuscript are marked in red and underline.

We will answer in a point-to-point fashion to the comments of the reviewer.

1. about CWI

Please explain the rationale for selecting the protocol of 15°C for 10 minutes.

⇒ Thanks for your comments. A review suggested that CWI could enhance recovery of performance in a variety of sports, with immersion in 10-15 °C water for 5-15 min duration appearing to be most effective at accelerating performance recovery (Versey et al., 2013). Excessive cooling via CWI may impair neuromuscular control by reducing nerve conduction velocity, balance, and joint stability during exercise [9], even increase the risk of injury. In our previous study, we observed that CWI at 15°C could improve the subsequence agility and sprint performance in the heat (Zhang et al., 2023). This suggests that CWI at 15°C may be an effective method of recovery and that it does not have a negative impact on neuromuscular control in the heat. Moreover, Brophy-Williams et al. (2011) found that CWI performed after a high intensity interval exercise resulted in better next day exercise performance. Consistent with Brophy-Williams et al. (2011), we selected the protocol of 15°C for 10 minutes in the present study. We have added these words in our manuscript.

2. interpretation of the effect of CWI

The reviewer agrees that CWI reduced the perception of unpleasant sensory stimuli associated with exercise-induced fatigue, but the reviewer has the impression that it is difficult to argue about fatigue recovery from this study. Rather, the reviewer is concerned that the reduction of unpleasant sensory stimuli may induce the false perception that fatigue is reduced, which in turn may induce overwork. Please clarify this point in the Discussion.

⇒ Thanks for your comments. In the manuscript, the term "fatigue recovery" is used to describe that CWI could promote recovery (etc. maintaining exercise capacities) via the inhibition of exercise fatigue. a systematic review with meta‑analysis and meta‑regression suggest that CWI was an effective recovery tool after high-intensity exercise, with positive outcomes occurring for muscular power and perceived recovery (Moore et al., 2022). McCarthy et al (2016) reported that 10 minutes of CWI at 8°C significantly reduced Tcore, HR, and RPE during subsequent exercise and improved subsequent endurance exercise performance. Consistently, we found that CWI could effectively reduce Tcore, HR, and RPE. Although we did not measure post-intervention physical performance in this study, we speculate that 10 minutes of CWI at 15°C also has the potential to aid recovery. The reduction in Tcore via CWI could increase global electroencephalographic β activity (and presumably overall α:β ratio) and decrease a sense of RPE. Therefore, CWI closely mirrors the extent of central fatigue during exercise, resulting in a more even pacing strategy so that higher power outputs can be better maintained during the subsequent exercise (De Pauw et al., 2013). Considering that 10 minutes of CWI at 15°C did not affect neuromuscular control or cause joint injury during subsequent exercise, it can conclude that 10 minutes of CWI at 15°C can be used after training or between competitions to promote recovery. Moreover, it is important to note that decreasing RPE through CWI may lead to a false perception of reduced fatigue, which in turn may induce overwork. We have added these words in our manuscript.  

Reviewer 3 Report

Comments and Suggestions for Authors

The article is well-written and demonstrates commendable originality. The clarity and precision in the writing are notable, and the innovative approach to the topic contributes significantly to the existing body of knowledge. However, to further enhance the manuscript, I recommend the following improvements:

1. **Improvement of Figures**: Consider revising the figures to ensure they are as clear and informative as possible. Each figure should effectively illustrate the points being made in the text and be of high graphical quality. This might involve redesigning them for better visual impact or clarifying the captions to improve reader understanding.

2. **Organization of Text and Figures**: Ensure that the organization of the text and figures is logical and intuitive. The flow from one section to another should be seamless, with figures appropriately placed to complement the text. This might mean rearranging some parts of the manuscript to align the figures more closely with the corresponding textual descriptions.

These enhancements will not only improve the readability and effectiveness of the paper but also emphasize the originality and impact of your research. By making these changes, the paper will likely be better received by the reviewing community and have a stronger overall presentation.

Author Response

        We appreciate the reviewer’s suggestions, which will help to increase the scientific quality of the paper and improve the resubmitted version. The corrections according to the comments in the revised manuscript are marked in red and underline. We will answer in a point-to-point fashion to the comments of the reviewer.

1. **Improvement of Figures**: Consider revising the figures to ensure they are as clear and informative as possible. Each figure should effectively illustrate the points being made in the text and be of high graphical quality. This might involve redesigning them for better visual impact or clarifying the captions to improve reader understanding.

Thanks for your comments. We revised our figures.

2. **Organization of Text and Figures**: Ensure that the organization of the text and figures is logical and intuitive. The flow from one section to another should be seamless, with figures appropriately placed to complement the text. This might mean rearranging some parts of the manuscript to align the figures more closely with the corresponding textual descriptions.

⇒ Thanks for your comments. I reorganized our Text and Figures.

Round 2

Reviewer 1 Report

Comments and Suggestions for Authors

Manuscript ID: brainsci-2987042 

Manuscript ID: brainsci-2987042 

The Effects of Post-Exercise Cold Water Immersion on Neuromuscular Control of Knee

The main message of the manuscript has not changed

The essential data upon while the conclusion were drawn have not changed.

Comments on the Quality of English Language

suffucuent